# Enabling Better Nutrition and Physical Activity for Adolescents from Middle Eastern Backgrounds: Focus Groups

**DOI:** 10.3390/nu13093007

**Published:** 2021-08-28

**Authors:** Nematullah Hayba, Claudia Khalil, Margaret Allman-Farinelli

**Affiliations:** Discipline of Nutrition and Dietetics, Charles Perkins Centre, Faculty of Medicine and Health, University of Sydney, Sydney 2006, Australia; ckha0190@uni.sydney.edu.au (C.K.); margaret.allman-farinelli@sydney.edu.au (M.A.-F.)

**Keywords:** adolescents, focus group, eating habits, nutrition, physical activity, enablers, barriers

## Abstract

The obesity epidemic in adolescents from Middle Eastern (ME) backgrounds necessitates co-designed and culturally-responsive interventions. This study’s objective was to gather the opinions, attitudes, capabilities, opportunities and motivations of ME adolescents residing in Australia on healthy eating and physical activity (PA) behaviours to inform a future prevention program. Five focus groups were conducted, with 32 ME participants, aged 13–18 years, recruited via purposive and snowball sampling. More participants were female (*n* = 19) and from lower socioeconomic areas (*n* = 25). A reflexive thematic analysis was performed using the Capability, Opportunity, Motivation-Behaviour (COM-B) model as the coding framework. Limited nutritional knowledge and cooking skills accompanied by a desire to make dietary changes were reported. Local and school facilities provided adolescents with PA opportunities, but participants declared safety concerns and limited opportunities for females and older grade students. Social support from family and friends were enablers for both healthy eating and PA. Cravings and desire for cultural foods influenced food choices. Individual and/or group approaches using social media and face-to-face format were recommended for future programs. To enable PA and dietary behaviour changes, interventions should be specifically tailored for ME adolescents to improve their nutrition literacy and skills, along with providing safe environments for sport in conjunction with social support.

## 1. Introduction

Overweight and obesity are major global public health concerns with rates continuing to rise among children and adolescents aged 5–19 years from 4% in 1975 to 18% in 2016 [1]. In Australia, the most recent comprehensive survey (2015) from the most populous state, New South Wales (NSW) [2], indicated that 27.4% of adolescents in secondary school aged 13 to 16 years were classified as either overweight or obese. Importantly, obesity is not shared equally with some ethnic groups more affected, and prevalence differentiated by gender and age. The prevalence of overweight and obesity was significantly higher among adolescents from Middle Eastern (ME) cultural backgrounds (41.1%) compared to those from English-speaking backgrounds (26.1%) [2]. This is of particular concern in ME boys of which 45.2% were overweight or obese compared to 35.9% of ME girls. In 2015, among ME boys in Year 8 (13–14 years), disturbingly, 64.9% were classified as overweight or obese compared with 25.6% among ME boys in Year 10 (15–16 years). This is of grave concern given the negative health impacts that arise from childhood obesity such as early adult onset of complications, cardiovascular diseases, insulin resistance, respiratory problems and psychiatric manifestations [1].

The survey also assessed the physical activity (PA) and dietary behaviours implicated in unhealthy weight gain, revealing adolescents from ME cultural backgrounds were less likely to meet targets in key areas [2] compared to those from English-speaking backgrounds. This included poorer cardiorespiratory fitness, with less adolescents reaching the Healthy Fitness Zone (39% vs. 61%), increased lack of awareness of recommended daily screen time limits (36% vs. 58%), more frequent eating in front of television (37% vs. 19%), drinking more than 1 cup of sugar-sweetened beverages per day (24% vs. 10%), more unrestricted snacking (73% vs. 50%) and being less likely to be meeting the recommended daily intake of vegetables (3% vs. 11%) [2]. Thus, the drivers for differences in obesity prevalence by ethnicity are plausibly environmental and behavioural, rather than due to genetic factors [2,3]. It seems apparent that health promotion is needed to address the specific needs of adolescents from ME backgrounds.

Systematic reviews [4,5] have revealed that most lifestyle interventions targeting adolescents are not effective and research concerning programs for adolescents from ethnic minority backgrounds (e.g., ME) is scarce. Studies to highlight inclusion of formative research and co-design with the target population are integral for developing effective interventions [6,7]. Thus, the aim of this research study was to gather the opinions, attitudes, capabilities, opportunities and motivations of adolescents aged 13–18 years from ME backgrounds residing in Australia on weight-related behaviours, to inform the content and delivery for a future prevention program.

## 2. Materials and Methods

This research study employed a qualitative research design. Focus groups were determined to be the best method to aid co-design of a future intervention program as it facilitates interaction allowing the exchange of ideas and experiences not usually possible for a one-to-one interview, stimulating rich data for analysis [8,9]. Peer influence is a significant determinant of healthy eating and physical activity during adolescence and having focus groups with peers was deemed important for exploration of these phenomena [10,11,12]. This qualitative research is reported according to the Consolidated Criteria for Reporting Qualitative Research (COREQ) [13]. This study was approved by the University of Sydney Human Research Ethics Committee (2021/154).

### 2.1. Recruitment

Eligible participants were identified using purposive and snowball sampling. To be eligible, participants met the following criteria: adolescents aged between 13–18 years, from Middle Eastern backgrounds, residing in Australia and English-speaking. Participants were recruited via posts on social media platforms (e.g., Instagram and Facebook), flyers and snowball method. A recruitment poster was posted on social media platforms, where existing connections and public groups within networks of the researchers were encouraged to share the post and inform others about the study to reach a wider audience. Flyers were posted on public notice boards in local government areas where people of ME backgrounds live or congregate. Flyers were also distributed in community facilities in these areas (e.g., recreational centres, shopping centres, libraries, local community hubs and parks). Participants were made aware that participation was voluntary.

Adolescents interested in the study were able to scan a quick response (QR) code on the flyer, which directed them to a short screening and demographic questionnaire on the Research Electronic Data Capture (REDCap, Vanderbilt University, Nashville, TN, USA, 2004). The questionnaire captured a participant’s presence of ME background, age, gender, email and postcode. Postcode was collected to determine the Socio-Economic Index for Areas (SEIFA) that is used in Australia to determine socio-economic status [14]. If they met the eligibility criteria, they were presented with the Participation Information Statement and Consent Form. Participants who provided e-consent were contacted by one researcher (NH) to agree on a date to attend the focus group.

### 2.2. Procedure and Data Collection

Focus group sessions were structured in accordance with the methodology suggested by Krueger and Casey [15] to achieve a robust discussion and adequate data collection. Sample size was based on satisfying the concept of information power which enabled researchers to obtain a range of opinions related to the topics being investigated [15,16]. Each focus group session aimed to run for approximately one hour and recruit more than six participants. Focus groups were conducted online via Zoom (due to COVID-19) by two female researchers, where researcher 1 (N.H., dietitian and PhD candidate) led and facilitated the discussion, while researcher 2 (C.K., student dietitian) recorded field notes such as nonverbal communication and group dynamics.

At the start of each session, five minutes were allocated for a brief introduction and building rapport. Participants were also informed that their personal data would be kept confidential. The experienced researcher facilitated a rich discussion using a set of questions developed *a priori* (Table 1) and structured around the Capability, Opportunity, Motivation-Behaviour (COM-B) model [17] to explore perceptions, barriers and facilitators to their capability, opportunity and motivation for physical activity, leisure screen time and healthy eating behaviours. The COM-B model is a behavioural system that describes the performance of a behaviour to be a result of the interplay between psychological capability, physical capability, social opportunity, physical opportunity, reflective motivation, and automatic motivation. Psychological capability refers to having the knowledge about nutrition and food and psychological skills such as reasoning, self-efficacy, and self-regulation to control your food and physical activity behaviours. Physical capability involves having the physical function to perform the behaviour such as being able to take part in exercise and sport. Reflective motivation refers to self-conscious evaluation and planning around eating and activity whereas automatic motivation denotes thinking linked to impulse, emotion, desires, and habits so that these are the default food and physical activity behaviours. Opportunity is classified as physical and social with physical involving non-living aspects of the environment such as food outlets in the local and school area, parks and cycle ways, and time such as material and financial resources to select the foods and participate in sports. Social opportunity encapsulates opportunity related to other people and possibly, organisations [17,18] such as friends you eat with and sporting organisations offering training and organised games. Participants were also asked about their preferences regarding content and delivery of future interventions. Probes were used to stimulate further discussion and elaborate on themes. Upon completion, participants were provided with an AUD 20 voucher that could be used in a supermarket/department store.

### 2.3. Data Analysis

Recorded data were transcribed verbatim using a two-step transcription process, with the second hearing to cross check between transcript and audio recording to ensure accuracy (C.K., N.H.). This ensured that all data and themes were captured. Transcripts were de-identified but not returned to participants for any further checking. Audio data files and transcriptions were stored on the institutional secure Research Data Store. Coding was constructed using a reflexive thematic analysis with a combination of deductive and inductive approaches [19,20,21,22]. Data were analysed through the following five steps: (1) Becoming familiar with the transcripts, (2) Constructing initial themes, (3) Coding the themes, (4) Reviewing and refining the themes, and (5) Summarising and displaying the data using the Framework Method [23,24]. To illustrate, transcripts were independently reviewed and analysed by two researchers (N.H., C.K.) following the Framework Method. The two researchers independently read the transcripts (Step 1). Each researcher then independently developed preliminary coding around the themes from the 6 subcomponents of the COM-B model (Step 2) [13]. This followed the deductive reasoning approach, and any independent new themes that emerged (inductive) that could not be classified were also included. From this, a coding framework based on the COM-B model was applied and data were assigned to the multiple themes and sub-themes (Step 3). After initial independent coding, all three researchers met, discussed disagreements concerning themes and agreed on the final refined themes. All data was coded using the final coding framework (Step 4), then summarised, and representative quotes were selected for display (Step 5).

## 3. Results

A total of 36 adolescents from ME backgrounds living in Australia met the selection criteria. However, only 32 (19 females) participated in the focus groups due to delayed responses or inconvenient session times. Consequently, five focus group sessions (three mixed-gender, one male-only and one female-only group) were conducted over an eight-week period. It was observed in the first focus group (mixed-gender) that the males dominated the discussion, so a decision was made to conduct same-sex groups to see if any different themes emerged. It was observed that the females were more open to discussion in the female-only group. This finding was not evident among males. However, no new themes emerged from the single versus group for both females and males. The size of the groups were 4 to 9 participants and only one had four (due to cancelations). The median duration of the groups was 56 min. The focus group duration ranged from 40 min to 1 h 19 min with 25 min an outlier for the small focus group (4 people). The majority of adolescents were between the age of 15–18 years (*n* = 21) and from low to middle socioeconomic (SES) neighbourhoods (*n* = 25) as determined by SEIFA. Demographic information of the sample is displayed in Table 2.

The findings are summarised below under five themes and supportive quotes are presented in Table 3. No discussion was coded to physical capability.

### 3.1. Psychological Capability

#### 3.1.1. Food Knowledge and Skills

The majority of participants reported no more than basic knowledge about healthy eating. However, adolescents showed an interest in receiving information on healthy eating from a trustworthy source, which includes information about the five food groups, healthy ME recipes, benefits of and strategies to follow a healthy lifestyle. The use of videos and simple nutrition tips as opposed to long texts were recommended, including simple cooking tips for recipes that would involve minimal cleaning and limited use of unfamiliar equipment and ingredients.

#### 3.1.2. Potential Interventions

Despite the abundance of commercial nutrition, cooking and PA applications (apps), respondents highlighted the need for trustworthy online apps that are specific for adolescents from ME backgrounds. Some were keen on cooking classes to facilitate independence from reliance on parents and others were interested in cultural foods to explore various cuisines. Older adolescent boys and girls engaged in PA stated that personalised advice (e.g., diet plan) and nutrition information specific to their sport would be beneficial and were interested in body image.

### 3.2. Social Opportunity

#### 3.2.1. Physical Activity

The majority of adolescents indicated friends positively influence them to participate in PA. Some adolescents reported that they feel more comfortable learning the sport if their friends or family are part of the team. Adolescent girls suggested that most of their screen time is spent talking with their friends which was exacerbated by the coronavirus pandemic as it limited opportunities for exercising and socialising and forced them to stay home and spend more time on screens. In contrast, adolescent boys highlighted friends indirectly help them to limit screen time as they are naturally more active and social when they are together. A single participant emphasised that older ME girls feel more comfortable with single-sex sport teams.

#### 3.2.2. Food Behaviours

Parental support was a common theme with younger adolescents reporting that their food choices are influenced by their mother. Influence of friends was also frequently reported whereby most adolescents indicated that their friends influence them to eat out and get fast foods, which they labeled as ‘convenient’ foods. Others stated that their friends could influence them positively, especially if they share similar health goals.

#### 3.2.3. Potential Interventions

Adolescents highlighted the importance of social support, rather than merely having access to nutrition information and learning individual skills. They indicated that group-based programs can provide them with social support and preferred small groups, as they can receive greater attention and gain a better understanding of the content.

In regards to delivery mode, there was no general consensus with some respondents suggesting that a face-to-face delivery mode allows them to interact with others, receive personalised advice and gain individual skills, while others stated that face-to-face gatherings can be confrontational, and the program schedule might not be appropriate for everyone. They recommended the use of social media for its convenience and connectivity with others. Participants concluded that a combination of social media, with face-to-face, individual and group-based approaches, would provide options and convenience.

Direct parental involvement was not favoured due to the generation gap, stating that having a set age group makes them more comfortable in sharing their perspectives. A few younger adolescents proposed indirect parental involvement, as they acknowledged their major role in supporting their children to achieve a healthy lifestyle.

School was preferred over a community-based approach as it enables them to be with their friends who could influence each other to participate in the program. Some believed it should be integrated into the school curriculum. Similarly, several adolescents expressed their interest in small group cooking classes with individual workstations. They indicated that they could support one another, as well as compete with each other, which makes the classes more fun, interactive and intriguing. Others lacked interest, relegating this responsibility to their parents.

### 3.3. Physical Opportunity

#### 3.3.1. Physical Activity

Adolescents highlighted that their school allows them to engage in different forms of PA, but a few younger adolescent girls identified homework load and inaccessibility of sports and PA facilities on campus to be barriers. Some participants declared that they also partake in sports outside of school. With regards to local facilities that promote PA, the safety of the area seemed to be a barrier. Younger adolescent girls suggested that despite having parks in their neighbourhoods, their parents feel it is unsafe to attend the park alone. This issue was mostly common among adolescents living in low to middle SES neighbourhoods. Adolescents also reported a lack of PA opportunities for girls and students in older grades. Participants were aware of their excessive screen time but reported a lack of alternative activities to replace it. Nevertheless, they stated schools help them limit the amount of screen time.

#### 3.3.2. Food Behaviours

Availability and accessibility of food in the house were common influences on eating behaviours and participants indicated the availability of certain snack foods in the house gave them the urge to consume them. The same adolescents who declared that their parents mostly prepare food at home, also reported that they mainly consume homemade foods. School and weekend times were when they were more likely to buy foods outside home which was generally fast food and more among boys than girls. Only younger adolescent girls commented on a potential barrier to cooking was their mothers’ beliefs that it was unsafe for children to cook.

#### 3.3.3. Potential Interventions

Adolescents discussed the physical online environment that might be suitable to support a healthy lifestyle program. Instagram was their social media program of choice and TikTok was also mentioned. Some younger adolescents only had access to YouTube.

### 3.4. Reflective Motivation

#### 3.4.1. Physical Activity

Participants acknowledged the importance of PA for their mental and physical health. The influence of social media celebrities on body image appeared to be a motivator for participating in PA, especially for girls. The presence of friends and family during sport games facilitated continuous participation. Despite the provision of PA opportunities by local and school facilities, a few older adolescents asserted that consistency with exercise is difficult due to schoolwork and other commitments. The discussions enabled reflection on alternative activities not mentally taxing for screen time reduction whilst some agreed that use differed according to intent such as mindless scrolling versus listening to a podcast.

#### 3.4.2. Food Behaviours

Participants agreed parental support or having health fitness goals regulated consumption of discretionary foods, despite adolescents enjoying eating them. Others reflected having an unhealthy diet as they regularly consume “junk foods”. A few participants noted their diet is high in ‘carbs’ (e.g., pasta, rice and bread), which they believed could lead to weight gain. Others reflected on the low consumption of vegetables, fruits and water and associated these habits with some health concerns.

Adolescent girls and boys stressed the necessity of maintaining a healthy diet to increase self-confidence and esteem and prevention of certain diseases. Participants, especially girls and boys who had fitness goals, expressed their interest in learning ways to improve their diet. They specified that information on portion sizes, meal timing, as well as strategies to reduce unhealthy food consumption and include more vegetables and fruits would be of benefit. Few adolescent boys were interested in diet changes despite their “unhealthy” diet and would only be concerned if they were playing in a representative sport team. They were aware that learning cooking techniques and cooking their own healthy dishes could motivate them to consume healthy food. They suggested that they will be mindful of the effort required to prepare the meal, including clean-up, and will have knowledge of the nutritional composition of the dish.

#### 3.4.3. Potential Interventions

Adolescents suggested that the ideal healthy lifestyle program must be fun, entertaining and eye-catching and agreed that social media or other online applications would be convenient due to their regular and widespread use. Some mentioned that a face-to-face approach is better for interaction, yet others emphasised that they feel more comfortable to socialise online. Participants suggested increased engagement in programs catered for ME cultural backgrounds as they enjoy their cultural foods and could relate to each other. Participants reiterated the importance of targeting the program at similar age groups (i.e., adolescents only) because of increased interrelatedness due to similar experiences, and preferred independence as healthy eating and PA are personal matters. However, they agreed that parents play a major role in influencing their food choices and suggested that indirect involvement (e.g., sending them emails) could help to support their health goals.

### 3.5. Automatic Motivation

#### 3.5.1. Physical Activity

Regarding PA, both genders stated that they enjoy playing sports such as soccer, basketball, Australian Rules Football, swimming and going to the gym, so participation was easy. Others indicated that PA is fun and it improves their mood, which motivates them to play sports. Adolescents viewed screen time as a break from the stress of school and studying, describing it as a ‘fun’ and ‘go to’ activity. One older adolescent described screen time as a distraction from her insomnia.

#### 3.5.2. Food Behaviours

Cultural foods were the most preferable food. Some respondents reported a desire for sweets, chips, burgers and soft drinks with one attributing it to stress. Fast food was identified by the same participant to be the easy choice when working whilst another claimed no desire for fast food, preferring home-cooked food. Food appearance was highlighted by some adolescents as the major contributor to their food choices. Another stated that craving affects their food selection.

## 4. Discussion

Findings suggest that healthy eating and PA behaviours of ME adolescents are influenced by nutritional knowledge and cooking skills, social and environmental influences especially from parents and friends that encourage healthy versus unhealthy eating and participation in sport, as well as emotional responses, desires, habits, and beliefs (motivation). Many stated they had inadequate food knowledge and cooking skills and expressed a desire for education and training. Clearly, mothers had a pivotal role in determining food consumed by ME adolescents as they were mostly responsible for meal preparation. Enjoyment of physical activity with others and eating cultural foods were powerful motivators for action. Findings from this study are in agreement with earlier studies, which identify determinants of adolescents’ food choices include food cravings, convenience, social support from friends and family, availability of nutritious versus less healthy foods at home, influence of social media on body image, as well as having health and fitness goals [25,26].

Previous research emphasises that friends can have positive and/or negative influence on dietary and PA behaviours of adolescents [27,28]. The lack of control over food choices when eating out with a large group of friends was reported as a barrier to healthy eating [27]. However, peers who share similar attitude towards healthy eating and PA were viewed as facilitators to dietary and PA behaviours [27,28]. This highlights that a group-based approach could increase the engagement and success rate of an intervention [27,28]. Other studies have suggested that some adolescents might have difficulty to participate in groups [29], which was also evident in this study. Hence, a combination of individual and group approaches is important to allow for a flexibility within interventions [29].

Considerable barriers to physical activity were raised with reported safety concerns in some neighbourhoods as a barrier to PA. Schoolwork in older grades was another barrier to PA, particularly among older adolescent girls. These findings are consistent with the study conducted in Los Angeles with low-income Latino and African-American adolescents [30]. Boys in this study were more likely to engage in PA than girls, which could be due to safety concerns or lack of girls-only sport teams as seen in the HERison project [31]. This could also be related to the gender-based role expectations where ME girls are believed to have more household tasks than ME boys [32] or to gender stereotyping, which enjoins boys to be muscular [33].

Girls were more likely to engage in healthy eating behaviours than boys, which could be due to self-perception and social expectations of girls’ body image [33]. These gender differences were reported in the NSW Survey that prompted this research [2]. Additionally, adolescents in the current study were aware of the excessive time that they spend on screens, yet girls indicated a lack of alternative activities to help them reduce their daily screen time. Previous studies conducted with Saudi and Kuwaiti adolescents highlight that girls spend more time on sedentary as opposed to physical activities [34,35]. These findings emphasise that gender-specific programs may need to be considered given the divergence of behavioural determinants. Boys may need more help with food and nutrition literacy for healthy eating, and girls may need more support for engagement in physical activity.

Participants in this study suggested that the availability of nutritious foods at home facilitate healthy eating behaviours, despite their desire for unhealthy food consumption. This is in agreement with previous literature, which highlights that providing parental support and a conducive home environment could induce weight-related behaviours [33]. Despite that, adolescents preferred age-specific interventions and indirect parental involvement, which is consistent with a previous American study [36]. Moreover, adolescents in this study reported a higher consumption of unhealthy foods when at school. This could be explained by the findings of a recent Australian study, declaring a lack of healthy food options in high schools [26]. Earlier studies have emphasised the importance for future prevention programs to collaborate with stakeholders within the community (i.e., ME community) to provide environmental and interpersonal support [33,37,38].

A few adolescents indicated that knowing the benefits of healthy eating (e.g., prevents certain diseases), motivate them to improve their dietary habits. However, the majority of adolescents reported that this knowledge does not influence their food choices, which corresponds to other findings [39]. The influence of social media celebrities on body image was reported as a key contributor to adolescents’ food choices, which has also been revealed in a previous literature [39]. This is concerning as adolescents’ perception of health can become distorted, depending on the type and source of online content [39]. Respondents also indicated that social media could encourage them to participate in PA, since they regularly access social media platforms. This idea has also been shared across literature [40]. On the other hand, earlier studies have suggested that adolescents preferred face-to-face communication with healthcare providers [41]. These findings are similar to the present study, where some adolescents expressed their concerns regarding the reduction of social interaction and credibility of online content. This emphasises the importance of developing programs that encompass both delivery modes, social media and face-to-face, consistent with other findings [29]. This can be achieved by providing adolescents with information on social media from trustworthy sources, as well as organising face-to-face cooking classes, as reinforced in similar studies [29,37]. Despite respondents preferring school over community-based interventions, a previous systematic review has suggested that interventions are more effective if they involve schools and communities and/or families [42]. Nevertheless, schools that focus more on students from ethnic minority backgrounds and differentiate them from other students could potentially increase the risk of cultural and racial discrimination [43].

## 5. Strengths and Limitations

This is the first qualitative study to report on the enablers and barriers to healthy eating and PA in adolescents from a minority group, Middle Eastern, in Australia. This research will inform the development of a prevention program to address the specific needs of this ethnic minority group. The program will centre on healthy eating and active living for ME adolescents to prevent overweight and obesity and is not targeted for the treatment of existing obesity. Opinions of all adolescents were sought and weight criteria were not applied for participation to avoid any weight stigmatisation [44]. We held mixed-gender as well as two single-sex groups as participants to see if different themes emerged when adolescents were not inhibited by the opposite sex. Despite the convenience of online focus groups, it was difficult to observe the non-verbal reactions of some participants due to having their camera off. Additionally, some groups were smaller than desirable because of drop-outs.

## 6. Conclusions

The findings support the development of a culturally-relevant program that capitalises on the ubiquitous use of social media amongst adolescents, to deliver information from credible sources (e.g., registered dietitians) in the form of short videos and simple tips. Moreover, small group cooking classes could be organised at community hubs or school kitchens to enable adolescents to interact with each other, develop individual skills and receive personalised advice, if appropriate. High schools that have a high percentage of ME students could endorse these initiatives and maximise opportunities for females to participate in female-only teams. Government initiatives are needed for policies and curriculum development to support continuous optimum dietary and physical activity behaviours amongst older adolescents. Parents can be indirectly involved to maintain effective stakeholder collaboration [23,27,33] whilst nurturing their adolescent’s growing independence. Findings from this qualitative ME-specific study cannot be generalised to other groups but the research methods used here and similarities to studies of other ME adolescent groups elsewhere in the world may help others in conducting formative research for program design.

## Figures and Tables

**Table 1 nutrients-13-03007-t001:** Questions used in focus groups to gather information on the capabilities, opportunities and motivations of adolescents on weight-related behaviours and preferences regarding content and delivery of interventions.

**Perceptions, Barriers, and Facilitators to Physical Activity and Leisure Screen Time**
What types of physical activity do you enjoy and participate in? How important is physical activity to you and what motivates you to join exercise and sporting activities? Do you think your local and school facilities and friends make it easier or harder to do physical activity? How much of your leisure time do you spend watching screens? Do you think that this amount of time is about right, too much or too little? How do your friends influence this?
**Food and Eating**
What foods do you like to eat? How much of what you eat comes from home and how much is bought outside? Do you think your diet is healthy and is there anything you would like to change? Do you need any information on food and cooking? Do you think your friends influence what you eat? Do you think eating a healthy diet is important, and would you be interested in learning ways to enjoy food and have a healthy diet?
**Opinions Regarding Content and Delivery of Intervention**
We’re thinking about developing a program to assist young people to achieve their lifestyle goals. How do you think we should deliver it to young people? Social media? Face-to-face? Video? Voice call? Individual or group? Where would you want the program to run? School, your local community or via a family-based approach? Do you think your parents should be involved in this program? To what extent? What do you think will be helpful to learn about? Would you like to learn how to cook other foods as part of the program?

**Table 2 nutrients-13-03007-t002:** Demographic information of focus group participants (*n* = 32) *.

Demographic Characteristics	Number of Participants (*n*)
**Age Group (years)**	
younger adolescents (13–14 years)	11
older adolescents (15–18 years)	21
**Gender**	
Girls	19
Boys	13
**SEIFA^a^ Rank within Australia—Decile** [14]	
<5	25
≥5	7

* *n* stands for sample size; ^a^ SEIFA: Socio-Economic Indexes for Areas (measure of socio-economic status) [14].

**Table 3 nutrients-13-03007-t003:** Exemplar quotes from adolescents by key theme.

**Psychological Capability**
“Knowing what you eat and what’s in it can affect your diet a lot of the time. Sometimes a lot of the food my mum makes, I have no idea what’s in it... I think it’s healthy at first and then I realise, maybe not. So, maybe having that knowledge about, you know, what’s good for you, how your diet can be controlled... is really good to know.”—P3 (F), FG2 “I think a video would be the most productive way, maybe headlines, like five simple tips to change in your lifestyle to lose weight.”—P1 (M), FG2 “… like teenagers, teenagers, some aren’t taught the skills by their parents, sometimes they have two full-time working parents or they have a lot of siblings and their parents can’t really make time to show them how to cook and what not... like so the cooking classes might be a great idea to give teenagers what they need in order to make nutritious meals.”—P8 (F), FG4 “Just certain types of diets for different sports and how they can help you… And also how to diet, like how to go about changing the way you eat, for example, if you have a competition on and you want to get leaner and get in better shape, how to eat in order to achieve that goal.” —P6 (M), FG3
**Social Opportunity**
“A lot of my screen time is spent, like on FaceTime with my friends, especially during the holidays, and I can’t really see them because they’re always busy.” —P1 (F), FG1 “I think sometimes it’s more like a mental mind game, because I think in school we always say like ‘Oh he got a Big Mac from Maccas’ and then the people get and bring it to school… you just feel left out and you feel like you need to buy it. So sometimes I can see it as being like feeling isolated… ”—P4, (M), FG4 “I feel like it would be better as a group, because if your friends with you as well, it’ll encourage you to keep on going and your friends will be by your side to support you on the way.” —P3 (F), FG5 “I feel combination of the two. So it can be like group-based, but there is an opportunity to kind of connect with whoever’s holding the course over one on one to kind of get more personalised advice.” —P4 (F), FG2 “I think maybe not like a call like this with our parents because… it could become awkward... But I think something like emailing them or notifying them, giving them reminders to kind of make sure we’re eating healthy. That could be a good way to involve them.” —P6 (M), FG3
**Physical Opportunity**
“Yes, same. My school has three big fields, but we’re not allowed to be on them lunch and recess either.” —P2, (F) FG5 “It just feels unsafe to go to the park... unless like we have a like a parent with us.” —P6 (F), FG1 “… we have the facilities, but I feel like they restrict us a bit because I go to an all-girls school… although they, like, vouch for equality and fairness and like being equal with boys… they don’t really allow us to do physical sports.”—P2 (F), FG4 “More like home foods... because you always have something there to eat. So, there’s no point of going out to eat.” —P1 (M), FG2 “I agree, I think Instagram probably because you can have more images and promote the website better… you can have links and like you can kind of make connections. And there’s also hashtags to help, like promote your brand… only like teens and early 20s that people really use it. And that’s kind of like… the target audience we’re aiming for.” —P2 (F), FG4 “… because if you have in the school curriculum… you’re forced to also like search up, look up different ingredients, recipes, maybe trainings that tell you how to eat properly.” — P2 (F), FG4
**Reflective Motivation**
“I saw a celebrity having like a nice body... But you can’t just get it. You have to, of course do engage in physical activity, eat well, it’s just, that’s how it is.” —P3 (F), FG1 “I think for me, it helps me deal with my anxiety, because as we’re getting like as I’m progressing with school and work is getting harder… I feel like just like a bit of jumping about or just playing about basketball with my brothers kind of just helps me ease my mind and stress. I feel like it’s a really good stress reliever. I agree.” —P2 (F), FG4 “… If you’re going to be a five hours on TikTok and social media is different to five hours teaching yourself something or reading an online book or listening to a podcast... ” —P3 (F), FG4 “Most of the food I eat is healthy, but there is still a lot of junk food in my diet and I’d like to cut it out.” —P3 (F), FG5 “Since I’m like Lebanese and... I like the food. So, I have like higher expectations. So, usually those like other recipes that are like low fat whatever, like they’re not usually that, something that I would like. If it’s made specifically for like, Middle Eastern kids, I would definitely like want to use that one more.” —P1 (F), FG1
**Automatic Motivation**
“Keeping healthy and training is good. It just makes you feel better.” —P1 (M), FG2 “If I see chocolate in front of me, I have to eat it.” —P3 (M), FG3 “Sometimes I can be a bit lazy and I’ll also just eat whatever is there... Sometimes I do feel like cooking but everything might be, like, a bit too complicated... so something, like, easy and... ingredients I can find in the pantry.” —P1 (F), FG1 “… I feel like that’s probably a really effective way to promote the program (cooking classes), because that way it’s you’re adding a bit of fun and you can have people work in like groups or even just by themselves or with an instructor on how to cook different things. So we’re broadening our knowledge as well, like having the experience and having fun.” —P2 (F), FG4

## Data Availability

Due to privacy reasons in ethics, we cannot make the data publicly available but please contact author if further information required.

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
