# Peer review of "Enabling Better Nutrition and Physical Activity for Adolescents from Middle Eastern Backgrounds: Focus Groups"

_nutrients, 2021, doi:10.3390/nu13093007_

Round 1

Reviewer 1 Report

Dear Authors, 

Thank you so much for preparing this very interesting and informative manuscript.  There is generally a lack of research in adolescents. 

I am happy with how methods and results are described. 

Author Response

Dear Reviewer,

Thank you very much for your comments. We are very pleased that you are happy with our manuscript.

Kind regards,

Nematullah Hayba, Claudia Khalil and Margaret-Allman-Farinelli

Reviewer 2 Report

The current manuscript seeks to present the perspectives and beliefs of adolescents from Middle Eastern backgrounds on the factors that impact their diet and physical activity. Additionally, the authors present recommendations for future interventions for these adolescents to improve their diet and physical activity levels. Adolescents from Middle Eastern backgrounds have a higher prevalence of overweight/obesity, poorer cardiorespiratory fitness, and worse diet-related behaviors. This is a need to better understand the drivers of these risk factors in this population, with the hope of addressing them via a culturally tailored intervention. The focus group methodology is appropriate. The results are well situated in the existing literature and the conclusions are supported by the results. However, there is a lack of clarity in the methods section and the results section. Adding detail to these sections would improve the interpretation of the results. There are also minor grammatical concerns throughout the manuscript that need to be addressed as well as other minor issues.

Major concerns:

  1. There is a lack of detail regarding the methods used during the focus groups. A set of questions were asked as shown in table 1. Then a discussion was had focusing on themes related to, “participants’ perceptions, knowledge, skills, motivations, and opportunities as potential barriers and enablers to physical activity, leisure screen time and healthy eating.” How was this discussion led? This is like a separate step in the data collection but there is also significant (but not complete) overlap with the questions presented in table 1. After this, participants were asked about their preferences for interventions. Is this the same as the “Opinions regarding content and delivery of intervention” section in table 1? It sounds both the same and separate. What there a reason for the closed-ended nature of the questions presented in table 1? Generally, it is considered good practice to ask open-ended questions in focus group settings. (What information do you need on food and cooking? Vs. Do you need any information on food and cooking?) Was this choice made for a reason? Also, how long was each group? Why did you allow participants to turn off their cameras? Did you have human subjects' approval for the research? (Major concern).

  1. The data analysis section lists 5 steps however the rest of the paragraph does not completely map on to these steps. For instance, what step is “After initial independent coding, all researchers met and agreed on a final coding frame for the qualitative data.”? It would be nice to have more detail for each of the steps. Maybe just a sentence or two about what was done. What is meant by becoming familiar with the transcripts? Etc… Also, list the 6 subcomponents of the capability-opportunity-motivation model that were used.

Minor concerns:

  1. The first paragraph presents a lot of valuable information. However, the presentation could be more clear. It is not clear which age group and years you're comparing in all the sentences. Are we looking at an increase over time? differences in gender? ethnic differences? Seems like everything is provided but without guiding the reader on what the intention of the sentence is.

  1. The second paragraph doesn’t present enough data. There is no comparison group presented or numbers provided.

  1. line 60. How were focus groups determined to be the best method? Citation?

  1. Why didn’t you have BMI inclusion criteria? If the hope is to create an intervention to address overweight/obesity in this population, why not target adolescents with overweight/obesity?

  1. line 100, explain what Cole’s is.

  1. line 130, the focus groups had between 4 to 9 participants. (not 6-9 and one group had 4).

  1. Table 2, it should be younger adolescents (13-15), older adolescents (15-18) according to the way you have age group (years). Also, 15 can’t be in both categories.

  1. What is SEIFA rank? Should be added to tables notes. It isn’t mentioned elsewhere in the manuscript, so not sure it is necessary.

  1. It would be helpful to have a brief definition of what is meant by each of the themes. Are these the COM-B model subcomponents?

  1. The discussion section has several instances where a citation is provided for what reads like a finding from the manuscript. (line 291 [19], line 311 [26], line 350 [24], line 352 [32,24], and line 311 [26]) Clarify what is from the manuscript and what the literature says.

  1. line 362-363, “We held mixed-gender as well as two single-sex groups as participants to see if differences emerged when adolescents were not inhibited by the opposite sex.” I didn’t see any analysis plan, results, or discussion addressing this.

Reviewer 3 Report

Thank you so much for your research work which could give many researcher in the field a deep understanding of the context of obesity prevention behaviours practices among adolescents.

I'd like to suggest a few minor revisions to make the manuscript more helpful for readers to understand better and develop further research.

  • It'd be better if quotation could be provided with each theme and category, rather than being presented in separate table.
  • The authors highlighted gender differences in factors affecting health behaviours, which are really important findings for development of prevention programs. However, suggestions on how different strategies should be applied in consideration of gender differences were not illustrated in detail.

Author Response

Dear Reviewer,

Thank you for comments. Please see your concerns addressed point by point below:

I'd like to suggest a few minor revisions to make the manuscript more helpful for readers to understand better and develop further research.

  • It'd be better if quotation could be provided with each theme and category, rather than being presented in separate table.
    • We do understand this could be helpful but we have several qualitative publications in a range of journals and the consensus is to include as a table as the best presentation. We have done this previously for Nutrients.
  • The authors highlighted gender differences in factors affecting health behaviours, which are really important findings for development of prevention programs. However, suggestions on how different strategies should be applied in consideration of gender differences were not illustrated in detail.
    • We have added some additional detail to the discussion to address this.

Kind regards,

Nematullah Hayba, Claudia Khalil and Margaret Allman-Farinelli

Round 2

Reviewer 2 Report

Line 37, grammatically incorrect.

Lines 41-42, what is the purpose/take away from this sentence. Do ME boys have a tendency to have a healthy BMI as they age? Are these 13-14-year-olds going to keep their 65% prevalence of overweight/obesity?

Line 51, change to “more frequent eating in from of the television (27% vs 19%)”.

Line 52-53, the drinking, snacking, and vegetable intake all need a direction. It seems more/greater drinking, more/greater unrestricted snacking, less/fewer meeting veggie intake would be the appropriate ones to use.

Line 111, COM-B model needs to be spelled out at first mention.

Line 141, take out “Firstly”

Line 142, take out “Secondly”

Line 146, “any new themes that emerged” not emerging.

Line 151, capitalize step

Line 161 discussion not discusiion

Line 164 in the not in te

Line 168, thanks for adding the median length. It would be nice to have the range.

Results, again it would be nice to have a brief definition of each of the themes. Psychological capability is defined as…. Social opportunity means… etc.

Line 391, should it be emphasizes not emphasises

Line 413-415, still reads as though the authors intended to compare the themes from different sex groups. However, the early statement says that you wanted to ensure that the sexes weren’t inhibited in expressing their views in mixed-sex company. If the authors didn’t separate the groups “to see if different themes emerged” then, take this language out. If they do want to see if different themes emerged, this information should be presented somewhere.
